# Structure–Activity Relationship Studies of Substitutions of Cationic Amino Acid Residues on Antimicrobial Peptides

**DOI:** 10.3390/antibiotics12010019

**Published:** 2022-12-23

**Authors:** Mayu Takada, Takahito Ito, Megumi Kurashima, Natsumi Matsunaga, Yosuke Demizu, Takashi Misawa

**Affiliations:** 1National Institute of Health Sciences, 3-25-26, Tonomachi, Kawasaki-ku, Kawasaki-shi, Kanagawa 210-9501, Japan; 2Graduate School of Medical Life Science, Yokohama City University, 1-7-29 Yokohama, Kanagawa 230-0045, Japan

**Keywords:** antimicrobial peptides (AMPs), cationic amino acid, secondary structure, chemical stability, digestive enzymes

## Abstract

Antimicrobial peptides (AMPs) have received considerable attention as next-generation drugs for infectious diseases. Amphipathicity and the formation of a stabilized secondary structure are required to exert their antimicrobial activity by insertion into the microbial membrane, resulting in lysis of the bacteria. We previously reported the development of a novel antimicrobial peptide, 17KKV, based on the Magainin 2 sequence. The peptide was obtained by increasing the amphipathicity due to the replacement of amino acid residues. Moreover, we studied the structural development of 17KKV and revealed that the secondary structural control of 17KKV by the introduction of non-proteinogenic amino acids such as α,α-disubstituted amino acids or side-chain stapling enhanced its antimicrobial activity. Among them, peptide **1**, which contains 2-aminobutyric acid residues in the 17KKV sequence, showed potent antimicrobial activity against multidrug-resistant *Pseudomonus aeruginosa* (MDRP) without significant hemolytic activity against human red blood cells. However, the effects of cationic amino acid substitutions on secondary structures and antimicrobial activity remain unclear. In this study, we designed and synthesized a series of peptide **1** by the replacement of Lys residues with several types of cationic amino acids and evaluated their secondary structures, antimicrobial activity, hemolytic activity, and resistance against digestive enzymes.

## 1. Introduction

Recently, several antibiotics with different mechanisms of action have been reported to save many lives from infectious diseases [1]. However, incorrect usage and long-term administration lead to the generation of multidrug resistant bacteria (MDRB) and raise problems in clinical practice [2]. Therefore, a novel treatment for MDRB infections is urgently needed. With this objective, antimicrobial peptides (AMPs) have been greatly studied as potential next-generation drugs against infectious diseases [3]. AMPs generally target the bacterial membrane and exhibit antimicrobial activity by lysing the microbial membranes; therefore, multidrug resistance is likely to be rare [4]. Two structural features play an important role in exerting antimicrobial activity [5,6]. These are (1) the formation of stabilized secondary structures and (2) amphipathicity, consisting of hydrophobic and cationic amino acids. Therefore, the enhancement of these properties could be a promising strategy for the development of novel AMPs. The structural development of AMPs based on structural control has been reported [7,8,9]. Gellman et al. designed and synthesized AMPs using cyclic β-amino acids and revealed that helical β-peptides showed potent antimicrobial activity and gained resistance against digestive enzymes [10,11]. Moreover, the introduction of side-chain stapling into magainin 2 (Mag2), which is a representative AMP, stabilizes its helical structure and exhibits potent antimicrobial activity even against MDRB in vivo [12]. Previously, we also reported the development of an AMP, 17KKV, based on the Mag2 sequence by increasing amphipathicity due to the replacement of amino acid residues [13]. We also studied the structural development of 17KKV and revealed that the secondary structural control of 17 KKV by the introduction of non-proteinogenic amino acids such as α,α-disubstituted amino acids or side-chain stapling enhanced their antimicrobial activity. Among these, peptide **1**, which contains 2-aminobutyric acid (U) residues in the 17KKV sequence, showed potent antimicrobial activity against multidrug-resistant *Pseudomonus aeruginosa* (MDRP) without significant hemolytic activity against human red blood cells [13]. Moreover, we investigated the effects of introducing side-chain stapling into the sequence on their antimicrobial and hemolytic activities [14].

As described above, AMPs have been structurally developed based on secondary structural control. In contrast, cationic amino acids play a pivotal role in exerting their antimicrobial activity by interacting with the microbial membrane. Therefore, further structural development based on the substitution of cationic amino acids is a promising strategy for the development of novel AMPs. Several AMPs possessing various cationic amino acids have been reported [15,16]. In this study, we designed and synthesized a series of Mag2 derivatives based on the substitution of cationic amino acid residues. Namely, the Lys residues of peptide **1** were replaced with other cationic amino acid residues such as diaminobutyric acid (Dab), Ornitine (Orn), Arg, and His, and their preferred secondary structures, antimicrobial activity, hemolytic activity, and chemical stability against proteolytic enzymes were evaluated (Figure 1). 

## 2. Results

### 2.1. Design and Synthesis of AMPs 

First, we designed and synthesized peptide **1** derivatives **2**–**5** by replacing Lys residues with several cationic amino acid residues, as shown in Table 1. Orn and Dab have propylamine and ethylamine moieties as side chains and have pKa values similar to those of Lys, while Arg and His have guanidino and imidazole groups, respectively. These side chains show differences in hydrophobicity and pKa [17] and may affect their secondary structures, antimicrobial activity, and chemical stability against digestive enzymes. The designed peptides were synthesized by solid-phase-peptide synthesis, purified by preparative high-performance liquid chromatography, and the target peptides were identified by mass spectrometry.

### 2.2. Preferred Secondary Structures of the Synthesized Peptides

Circular dichroism (CD) spectra analysis of the synthesized peptides was performed in phosphate buffer with 1% sodium dodecyl sulfate (SDS) solution to investigate the effects of replacement of Lys residues with other cationic amino acids on their secondary structure. As shown in Figure 2, all peptides showed negative maxima at approximately 208 and 222 nm, indicating that the peptides formed helical structures [18]. These results suggested that the replacement of cationic amino acids did not affect the helical structure of Mag2. The α-helix content was calculated from the CD spectra of each tested peptide. The substitution of Lys residues with Orn did not affect the α-helix content, whereas the replacement of Lys with Dab or His significantly decreased the α-helix content. Cationic amino acids and Phenylalanine form cation-π interactions, resulting in the stabilization of its helical structure. However, it has been reported that Dab has a weaker interaction with Phe residues than with Lys or Orn residues because of its shorter alkyl chain [19]. On the other hand, His residues tend to be epimerized and could disrupt their secondary structure [20]; therefore, we concluded that the introduction of Dab and His residues decreases the helix content.

### 2.3. Antimicrobial Activity and Hemolytic Activity

Next, the antimicrobial and hemolytic activities of the synthesized peptides against human red blood cells were evaluated. The minimum inhibitory concentration (MIC) was measured against both Gram-positive and Gram-negative bacteria, and hemolytic activity was evaluated by measuring the 535 nm absorption derived from the leakage of hemoglobin. The inhibitory activities of the synthesized Mag2 derivatives are shown in Table 2. It was found that peptides **2** and **3**, which have Orn and Dab, showed the same extent of antimicrobial activity against Gram-positive and Gram-negative bacteria as peptide **1**. These results indicate that the Lys residues of peptide **1** could be replaced with Orn and Dab. In contrast, peptides **3** and **4** showed weak antimicrobial activity compared with peptide **1**, and the Arg-substituted derivative showed potent hemolytic activity. Cell-penetrating peptides (CPPs) are mainly composed of Arg residues, which play an important role in cell membrane permeability [21]. Therefore, the substitution of Lys residues with Arg enhances the interaction with the human cell membrane and induces cytotoxicity against human red blood cells. Peptide **5** showed the lowest antimicrobial activity among the tested peptides. His residues tend to be epimerized [20], and the introduction of His residues decreased the content of α-helix, resulting in a decrease in antimicrobial activity.

### 2.4. Chemical Stability of the Peptides against Digestive Enzymes

Finally, we investigated the chemical stability of the synthesized peptides **1**–**3** against digestive enzymes such as peptidases and proteases. Previously, it was reported that the introduction of Orn residues into the sequence increased the stability against serum, because the Orn residues are non-proteinogenic amino acids and are difficult to detect by digestive enzymes [22]. Based on this knowledge, the investigation of the chemical stability of peptides **2** was used to verify their utility as AMPs. The peptides were treated with proteinase K, which is a representative digestive enzyme and widely used for evaluation of chemical stability of peptides [23,24], and the amount of the remaining peptides was evaluated by high performance liquid chromatography (HPLC) analysis. As shown in Figure 3, peptides **2** remained at 41% after 24 h treatment with proteinase K, while peptide **1** was degraded to approximately 20%. These results indicate that the substitution of Lys residues with Orn residues increases chemical stability against proteinase K.

## 3. Discussion

In this study, we designed and synthesized a series of novel AMPs based on peptide **1**, which was developed in our previous study, by replacing Lys residues with other cationic amino acids, Orn, Dab, Arg, and His. The preferred secondary structure of synthesized peptides **1**–**5** was investigated. As shown in Figure 2, peptide **1**–**5** adopted a stabilized helical structure through the introduction of two Aib residues into the sequence. It has been demonstrated that replacement of Lys residues with other cationic amino acids has no significant effect on their helical structure. The antimicrobial and hemolytic activities of the synthesized peptides were also evaluated. It has been revealed that peptides **2**, **3**, and **4**, which have Orn, Dab, and Arg residues, respectively, showed the same extent of antimicrobial activity against Gram-positive and Gram-negative bacteria as peptide **1**. In contrast, peptide **5**, which has His residues, showed no or weak antimicrobial activity. Because AMPs target bacterial membranes, they also affect human cell membranes. Therefore, we evaluated the hemolytic activity of the synthesized peptides against human red blood cells. Generally, an increase in the hydrophobicity of amphipathic peptides tends to enhance their hemolytic activity [25]. Based on this knowledge, it was expected that peptides **2** and **3** would exhibit lower hemolytic activity due to their shorter alkyl side chains. Surprisingly, it was observed that the peptides **2** and **3** with shorter alkyl chain exhibited higher hemolytic activity than peptide **1**. It remains unclear why peptide **3** exhibited stronger hemolytic activity than peptide **1**, which may affect its affinity for the human membrane. Finally, the chemical stability of peptides **1** and **2** was investigated to confirm the utility of substituting Lys residues with Orn residues. Although peptide-based drugs show excellent efficacy, one of the challenges is that they are easily degraded by digestive enzymes [24]. Orn is a non-proteinogenic amino acid that is difficult to recognize for digestive enzymes and increases chemical stability of Orn-containing peptides. As shown in Figure 3, peptide **2** exhibited a longer t_1/2_ than peptide **1**, indicating that the replacement of Lys residues with Orn residues increased the chemical stability without any significant effects on their secondary structure, antimicrobial activity, and hemolytic activity. These results suggest that replacement of Lys residues with Orn residues could be a promising strategy for the development of AMPs.

## 4. Materials and Methods

### 4.1. General Information

Chemicals were purchased from Sigma-Aldrich Co. (St. Louis, MO, USA), LLC (Kanto Chemicals Co.). Inc. (Tokyo, Japan), Tokyo Chemical Industry Co. Ltd. (Tokyo, Japan), Wako Pure Chemical Industries Ltd. (Tokyo, Japan), and Watanabe Chemical Industries Ltd. (Hiroshima, Japan) and used without further purification. Mass spectra were obtained using a Shimadzu IT-TOF MS (Kyoto, Japan) instrument equipped with an electrospray ionization source.

### 4.2. Peptide Synthesis

The designed peptides were synthesized using Fmoc-based solid-phase methods. A representative coupling and deprotection cycle is described as follows: NovaPEG Rink amide resin was soaked for 30 min in CH_2_Cl_2_. After the resin was washed with DMF, Fmoc-amino acid (4 equiv.) and Hexafluorophosphate benzotriazole tetramethyl uronium (HBTU; 4 equiv.) dissolved in N-methyl-2-pyrrolidone (NMP) were added to the resin. DIPEA (4 equiv) and 0.1 M 1-hydroxynbenzotriazole (HOBt) in N-methyl-2-pyrrolidone (NMP) were added to the coupling reaction. Fmoc-protective groups were deprotected using 20% piperidine in DMF. The resin was suspended in a cleavage cocktail (95% TFA, water 2.5%, 2.5% triisopropylsilane) at room temperature for 3 h. The TFA was evaporated to a small volume under a stream of N_2_ and dripped into cold ether to precipitate the peptide. The peptides were dissolved in DMSO and purified by reverse-phase HPLC using a Discovery^®^ BIO Wide Pore C18 column (25 cm × 21.2 mm solvent A: 0.1% TFA/water, solvent B: 0.1% TFA/MeCN, flow rate: 10.0 mL·mL^–1^, gradient: 10–90% gradient of solvent B over 30 min). After purification, the peptide solution was lyophilized. Peptide purity was assessed using analytical HPLC and Inertsil WP300 C18 5 µM 150 × 3.0 mm (25 cm × 4.6 mm; solvent A: 0.1% TFA/water gradient B: 0.1% TFA/MeCN, flow rate: 1.0 mL·mL^–1^, gradient: 10–90% gradient of solvent B over 30 min).

### 4.3. CD Spectrometry

The CD spectra were recorded using a 1.0 mm path length cell. The data are expressed in terms of [*θ*], that is, the total molar ellipticity (deg cm^2^ dmol^−1^). The peptides were dissolved in 20 mM phosphate-buffered solution (pH 7.4) with 1% SDS at a concentration of 100 µM.

### 4.4. Antimicrobial Activity

The selected bacterial strains were obtained from the Biological Resource Center, NITE (NBRC; Tokyo, Japan). *Escherichia coli* DH5α cells were purchased from BioDynamics Laboratory, Inc. (Tokyo, Japan). The antimicrobial activities of the peptides against two Gram-positive bacteria and three Gram-negative bacteria, *S. aureus* NBRC 13276, *E. coli* DH5α, *P. aeruginosa* NBRC 13275, and *MDRP* were measured using the standard broth microdilution method, as previously described. Briefly, the bacteria were inoculated and grown overnight at 37 °C on the media for other organisms i (agar medium) and then collected with the media for other organisms ii (liquid medium), according to the Japanese Pharmacopoeia 17th Edition. Each peptide was 2-fold serially diluted with an initial concentration of 4000 μM–1.56 μM in PBS for use. Next, 10 μL of the peptide solution per well was added to each well of a sterile 96-well plate. Subsequently, 90 μL per well of inoculation with 10^4^ CFUs (colony forming units) per mL was added to each well, and the plate was incubated for 18 h at 35 °C. MIC was defined as the lowest concentration of the peptide that completely inhibited bacterial growth by visual inspection at 535 nm.

### 4.5. Digestion Assay

The digestion assay was performed using Proteinase K of the digestive enzyme. Peptides (1 mM) were treated with 0.002% (*v*/*v*) Proteinase K and peptides were incubated at 37 °C for 2–24 h. A solution (0.1% TFA/H_2_O/MeCN) was added to quench the enzymatic reaction. The peptides were evaluated using analytical HPLC and a Discovery^®^ BIO Wide Pore C18 column (25 cm × 4.6 mm, solvent A: 0.1% TFA/water, solvent B: 0.1% TFA/MeCN, flow rate: 1.0 mL mL^−1^, gradient: 10–60% gradient of solvent B over 30 min). Reproducibility was confirmed by the two independent experiments.

### 4.6. Hemolysis Activity

Human red blood cells (RBCs) were kindly supplied by the Japanese Red Cross Society (Tokyo, Japan) and were collected from volunteers under informed consent. The hemolysis test of peptides was performed using a previously reported method [13]. In short, the RBCs were washed three times with 172 mM Tris-HCl buffer (pH = 7.6, wash buffer: WB), and 1.0 mL of sedimented erythrocytes was then transferred to 9.0 mL of WB. Then, 0.5 mL of this suspension was diluted in WB, and 50 µL of RBC suspension was incubated with 50 µL of each peptide for 30 min at 37 °C (final peptide conc from 100 to 0.39 μM). The suspensions were then centrifuged for 5 min at 400 rpm. The absorbance of the supernatant was measured at a wavelength of 535 nm. M-Lycotoxin [26] was used as a positive control, and the absorbances of the sample by treatment of DMSO and M-Lycotoxin were defined as 0 and 100%. Concentrations that showed hemolytic activity greater than 50% were defined as a hemolysis activity of each peptide.

## 5. Conclusions

In conclusion, we identified novel antimicrobial peptide **2** by replacing Lys residues with Orn residues based on peptide **1**. Peptide **2** adopts a stabilized helical structure and exhibits antimicrobial activity against both Gram-positive and Gram-negative bacteria. Peptide **2** also exhibits antimicrobial activity against MDRP without any significant hemolytic activity. Furthermore, replacing Lys with Orn increased the chemical stability against proteinase K and prolonged the half life time (t_1/2_) by up to approximately two times. We expect that peptide **2** will be a promising reagent for treatment of infectious diseases.

## Figures and Tables

**Figure 1 antibiotics-12-00019-f001:**
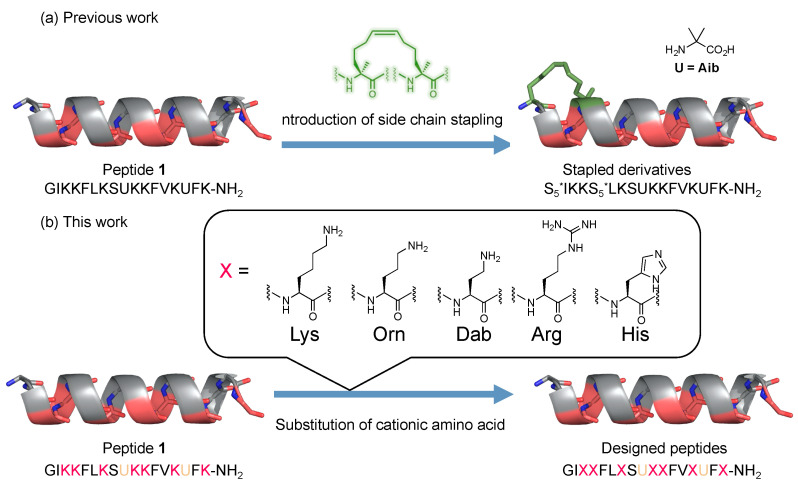
Design of a series of antimicrobial peptides based on the substitution of Lys residues with other cationic amino acids. (**a**) Previous works for the development of antimicrobial peptides based on the secondary structural control. (**b**) Investigation of the substitution of cationic amino acid residues in this study. The cyclization between *i* and *i + 4* of two S_5_ residues is indicated by an asterisk.

**Figure 2 antibiotics-12-00019-f002:**
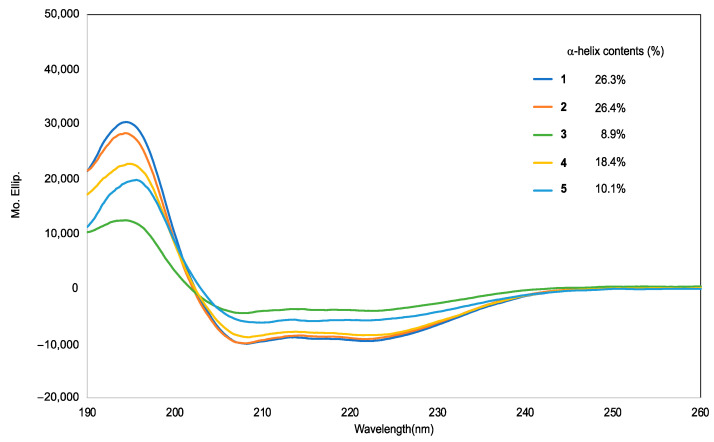
Secondary structural analysis of peptides **1**–**5** using CD spectra in pH = 7.4 phosphate buffer with 1% SDS.

**Figure 3 antibiotics-12-00019-f003:**
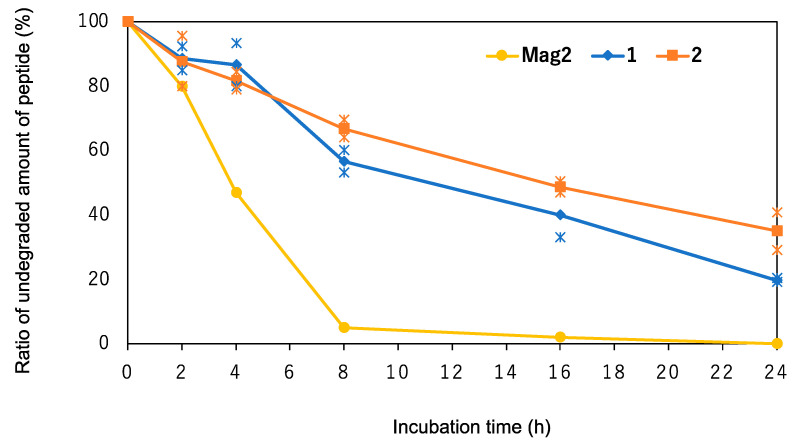
The chemical stability of Mag2 and peptides **1**–**2** against proteinase K. The peptides were treated with proteinase K (0.002% *v*/*v*) for indicating time (h), and the amount of undegraded peptides was measured by HPLC. The digestion assay were performed in two independent experiments and each data were shown as asterisks. The data of Mag2 was taken from ref [13]. Peptide concentration: 1 mM.

**Table 1 antibiotics-12-00019-t001:** The sequence of designed peptides and the protonated pKa values of side chains [17].

Peptide	Sequence	
Mag2	H-GIGKFLHSAKKFGKAFVGEIMNS-NH_2_	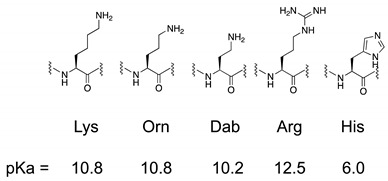
**1**	H-GIKKFLKSXKKFVKXFK-NH_2_	
**2**	H-GIOOFLOSUOOFVOUFO-NH_2_	
**3**	H-GIBBFLBSUBBFVBUFB-NH_2_	
**4**	H-GIRRFLRSURRFVRUFR-NH_2_	
**5**	H-GIHHFLHSUHHFVHUFH-NH_2_	

**Table 2 antibiotics-12-00019-t002:** The antimicrobial activity and hemolytic activity of Mag2 and peptides **1**–**5**. The minimum inhibitory concentration (MIC) was used for the evaluation of antimicrobial activity against four bacterial strains. *E.coli: Escherichia coli; P.a.: Pseudomonas aeruginosa: MDRP; multidrug-resistant Pseudomonas aeruginosa; S.a.: Staphylococcus aureus*. The antimicrobial and hemolytic assay were performed at the final concentration at from 0.39 to 100 μM. In the hemolysis assay, DMSO and M-lycotoxin were used as negative and positive control, and their hemolysis ratio were defined as 0 and 100%. Concentrations that showed hemolytic activity greater than 50% were defined as a hemolysis activity of each peptide.

Peptide	Sequence	MIC (µM)				Hemolysis (µM)
		Gram (–)			Gram (+)	
		*E.Coli*	*P.a.*	*MDRP*	*S.a.*	
Mag2	H-GIGKFLHSAKKFGKAFVGEIMNS-NH_2_	3.13	12.5	12.5	100	>100
**1**	H-GIKKFLKSXKKFVKXFK-NH_2_	3.13	3.13	3.13	12.5	100
**2**	H-GIOOFLOSUOOFVOUFO-NH_2_	3.13	6.25	3.13	12.5	50
**3**	H-GIBBFLBSUBBFVBUFB-NH_2_	3.13	6.25	6.25	12.5	25
**4**	H-GIRRFLRSURRFVRUFR-NH_2_	3.13	12.5	12.5	3.13	6.25
**5**	H-GIHHFLHSUHHFVHUFH-NH_2_	>50	>50	>50	>50	>100

## Data Availability

Data are available from the authors on reasonable request. HPLC analytical method and Characterization of peptides are in the Appendix A.

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
