# Peer review of "Structure–Activity Relationship Studies of Substitutions of Cationic Amino Acid Residues on Antimicrobial Peptides"

_antibiotics, 2022, doi:10.3390/antibiotics12010019_

Round 1

Reviewer 1 Report

The authors present another in a series of papers regarding modifications of antimicrobial peptides to improve on their efficacy. With some exceptions, the paper is well written and the English is of good quality. There are several points that ought to be addressed.

Line 38: "no possibility of generating MDRB".  This statement cannot be supported by evidence. Microbes are particularly gifted at developing resistance given sufficient selective pressure, and if the drugs are sufficiently specific to damage microbial membranes without damaging RBC membranes, resistance is conceivable. I humbly request that the authors use words more in line with the reference they cite and indicate that "multidrug resistance is likely to be rare."

Line 114: The wording "leakage of blood contents" is a little strange. Leakage of RBC cytoplasmic contents or hemoglobin would be clearer and more accurate.

Line 145: In Figure 3 I would much prefer to see a proportional X axis. The type of X axis used makes it very difficult to evaluate the rate of degradation of the different peptides. In my opinion, this type of X-axis differs little from a histogram or table and (see below) is not as helpful in evaluating stability half life. There is also no indication how many replications of each trial were done and no statistics. There are also no positive or negative controls. While I am not sure what the most appropriate controls would be, I feel confident that the authors, working in this field, would be able to identify appropriate controls.

Line 238: Should read "RBC suspension". Particulates (such as cells) in a solvent is a suspension, not a solution. I appreciate that the assay has been described previously, but there is a lot of information missing in the description of the hemolysis assay. The volumes of RBC suspension and peptide solution are of little value without knowing what the concentrations are. Likewise Table 2 is unclear; the extent of hemolysis (compared to controls) can be anywhere from 0 to 100%. If Table 2 (under "hemolysis") lists peptide concentrations, they are the concentrations that do what? That produce a result not statistically different from zero? Please clarify how the assay is being done and how the results are being presented.

Line 248: It is difficult to evaluate the statement about half life in the presence of proteinase K without a more user friendly type of graph and evidence of replicate values. Making the above recommended changes (at line 148) would strengthen this conclusion about half life. My understanding is that protease K is chosen as a broad spectrum protease, but I wonder if the authors could include a reference (only a suggestion) as to the suitability of protease K as a representative of human digestive enzymes, as presumably this would be an orally adminstered drug, thus the protease resistance testing?

The last line seems to have a number with no reference.

Author Response

Reviewer 1

  1. Line 38: "no possibility of generating MDRB".  This statement cannot be supported by evidence. Microbes are particularly gifted at developing resistance given sufficient selective pressure, and if the drugs are sufficiently specific to damage microbial membranes without damaging RBC membranes, resistance is conceivable. I humbly request that the authors use words more in line with the reference they cite and indicate that "multidrug resistance is likely to be rare."

Answer:

In the accordance with the reviewer’s comment, we corrected sentence in line 38, as follows;

From “no possibility to generate MDRBs”

To “multidrug resistance is likely to be rare”

  1. Line 114: The wording "leakage of blood contents" is a little strange. Leakage of RBC cytoplasmic contents or hemoglobin would be clearer and more accurate.

Answer:

In the accordance with the reviewer’s comment, we corrected sentence in line 113, as follows;

From “leakage of blood contents”

To “Leakage of RBC cytoplasmic contents or hemoglobin”

  1. Line 145: In Figure 3 I would much prefer to see a proportional X axis. The type of X axis used makes it very difficult to evaluate the rate of degradation of the different peptides. In my opinion, this type of X-axis differs little from a histogram or table and (see below) is not as helpful in evaluating stability half life. There is also no indication how many replications of each trial were done and no statistics. There are also no positive or negative controls. While I am not sure what the most appropriate controls would be, I feel confident that the authors, working in this field, would be able to identify appropriate controls.

Answer:

In the accordance with the reviewer’s comment, we corrected the X axis of Figure 3 as below. We also added the data of the digestion assay of Magainin 2 as a control. Moreover, we described the assay condition and the replications in line 147.

Figure 3. The chemical stability of Mag2 and peptides 1-3 against proteinase K. The peptides were treated with proteinase K (0.002% v/v) for indicating time (h), and the amount of undegraded peptides was measured by HPLC. The digestion assay were performed in two independent experiments. Peptide concentration: 1 mM.

  1. Line 238: Should read "RBC suspension". Particulates (such as cells) in a solvent is a suspension, not a solution. I appreciate that the assay has been described previously, but there is a lot of information missing in the description of the hemolysis assay. The volumes of RBC suspension and peptide solution are of little value without knowing what the concentrations are. Likewise Table 2 is unclear; the extent of hemolysis (compared to controls) can be anywhere from 0 to 100%. If Table 2 (under "hemolysis") lists peptide concentrations, they are the concentrations that do what? That produce a result not statistically different from zero? Please clarify how the assay is being done and how the results are being presented.

In the accordance with reviewer’s comment, the assay condition of hemolytic activity was added in experimental section as below:

In short, the RBCs were washed three times with 172 mM Tris-HCl buffer (pH = 7.6, wash buffer : WB) and 1.0 mL of sedimented erythrocytes was then transferred to 9.0 mL of WB. Then, 0.5 mL of this supension was diluted in WB, and 50 µL of RBC solution was incubated with 50 µL of each peptide for 30 min at 37 °C (final peptide conc from 100 to 0.39 mM). The suspensions were then centrifuged for 5 min at 400 rpm. The absorbance of the supernatant was measured at a wavelength of 535 nm. M-Lycotoxin [26] was used as a positive control and the absorbance of the sample by treatment of DMSO and M-Lycotoxin were defined as 0 and 100%.

  1. It is difficult to evaluate the statement about half life in the presence of proteinase K without a more user friendly type of graph and evidence of replicate values. Making the above recommended changes (at line 148) would strengthen this conclusion about half life. My understanding is that protease K is chosen as a broad spectrum protease, but I wonder if the authors could include a reference (only a suggestion) as to the suitability of protease K as a representative of human digestive enzymes, as presumably this would be an orally adminstered drug, thus the protease resistance testing?

Answer:

As reviewer’s comment, proteinase K is widely used as a representative protease for the evaluation of the chemical stability of peptides because of its broad protease spectrum. Therefore, we chose the proteinase K for the digestive assay. In the accordance with reviewer’s suggestion, we added the sentence and reference to explain why we chose the proteinase K In line 141.

“proteinase K, which is a representative digestive enzyme and widely used for evaluation of chemical stability of peptides [23-24],”

  1. The last line seems to have a number with no reference.

Answer:

In the accordance with the reviewer’s comment, the number in reference was deleted.

End

Reviewer 2 Report

Please refer to the question in the attachment.

Author Response

Reviewer 2

  1. Why using whole lysine substitution in peptide rather than point mutation?

Answer:

As the reviewer’s comment, the point mutation is proper to demonstrate the effects of the substitution of each cationic amino acid residue on their antimicrobial and hemolytic activity. On the other hand, the lysine and Ornitine have similar chemical structure and pKa value, thereby we hypothesized that the impacts of the point mutation could be small. Therefore, we designed the whole lysine substitution analogues.

  1. Orn and Dab substitution showed great potencies against Gram +/- as well as MDRP. Arg shows toxicity and less potency but suspected to increase the cell penetration. My question is: Do you think you could hybridize Orn, Dab and Arg to create a peptide with better potency and permeability? (also correlate my question in 2.1)

Answer:

As reviewer’s comment, the structure-activity relationship by the substitution of the cationic amino acid residues on antimicrobial and hemolytic activity, and chemical stability were demonstrated in this study. The combination of several cationic amino acid and introduction of side-chain stapling as previous study could increase the antimicrobial activity without any significant hemolytic activity. Further structural developments are under investigation, and their results would be published in near future.

  1. Is the proteinase K the most common one found in bacteria? Please support it by citing literatures.

Answer:

Proteinase K is representative protease and widely used for evaluation of the chemical stability of peptides. In the accordance with reviewer’s comment, the sentence and references were added in line 141 as below;

“proteinase K, which is a representative digestive enzyme and widely used for evaluation of chemical stability of peptides [23-24],”

End

Round 2

Reviewer 1 Report

Thank you for your improvements to this manuscript. There are just a couple of simple but important corrections that still need to be made.

Line 114: This reviewer apologizes for the ambiguity. Either “cytoplasmic contents” or “hemoglobin” would be suitable, not both.

Line 130: More clarity is still required for the legend of Table 2 and for Methods (Line 250). It is clear that the concentrations were varied from 0.39 to 100 mM in arriving at the MIC values. However, the different peptide concentrations would be expected to give a range of different % hemolysis values. If the list of values in the “Hemolysis” column are actually in micromolar as indicated in the table, what amount of hemolysis do they represent? Line 254 in Methods suggests that these are actually meant to be % hemolysis and not concentrations, in which case there is no indication of what peptide concentration (something between 0.39 and 100) produces that amount of hemolysis. Please provide more clarity.

Line 155-158: This reviewer greatly appreciates the change in graph type and the confirmation that the experiments were performed more than once. However, it seems highly unlikely that every single data point was exactly the same in both experiments. It would be inappropriate to graph the averages of two experiments, as this hides any variability in the measurements from view. It should be possible to put symbols from both results as well as the averages on the graph, place lines through the averages, and then hide the symbols for the averages, leaving just the actual data points with lines through the averages. This would allow the readers to draw their own conclusions about the reproducibility of the data and the amount of difference among the results for the various peptides. Alternatively, error bars could be used.

Line 249: still needs correcting: please change “RBCs solution” to “RBCs suspension”.  

Author Response

Line 114: This reviewer apologizes for the ambiguity. Either “cytoplasmic contents” or “hemoglobin” would be suitable, not both.

Answer:

In the accordance with reviewer’s comment, the sentence was corrected in line 114 as follows;

From “cytoplasmic contents or hemoglobin”

To “hemoglobin”

Line 130: More clarity is still required for the legend of Table 2 and for Methods (Line 250). It is clear that the concentrations were varied from 0.39 to 100 mM in arriving at the MIC values. However, the different peptide concentrations would be expected to give a range of different % hemolysis values. If the list of values in the “Hemolysis” column are actually in micromolar as indicated in the table, what amount of hemolysis do they represent? Line 254 in Methods suggests that these are actually meant to be % hemolysis and not concentrations, in which case there is no indication of what peptide concentration (something between 0.39 and 100) produces that amount of hemolysis. Please provide more clarity.

Answer:

In the accordance with reviewer’s comment, the authors added the explanation of antimicrobial assay and hemolysis assay in legend of table 2 and experimental section as shown below.

Table 2. The antimicrobial activity and hemolytic activity of Mag2 and peptides 1-5. The minimum inhibitory concentration (MIC) was used for the evaluation of antimicrobial activity against four bacterial strains. E.coli: Escherichia coli; P.a.: Pseudomonas aeruginosa: MDRP; Multidrug resistant Pseudomonas aeruginosa; S.a.: Staphylococcus aureus. The antimicrobial and hemolytic assay were performed at the final concentration at from 0.39 to 100 mM. In the hemolysis assay, DMSO and M-lycotoxin were used as negative and positive control, and their hemolysis ratio were defined as 0 and 100%. Concentrations that showed hemolytic activity greater than 50% were defined as a hemolysis activity of each peptide.

Line 155-158: This reviewer greatly appreciates the change in graph type and the confirmation that the experiments were performed more than once. However, it seems highly unlikely that every single data point was exactly the same in both experiments. It would be inappropriate to graph the averages of two experiments, as this hides any variability in the measurements from view. It should be possible to put symbols from both results as well as the averages on the graph, place lines through the averages, and then hide the symbols for the averages, leaving just the actual data points with lines through the averages. This would allow the readers to draw their own conclusions about the reproducibility of the data and the amount of difference among the results for the various peptides. Alternatively, error bars could be used.

Answer:

In the accordance with reviewer’s comment, the data of each experiment were added in Figure 3 as shown below:

Figure 3. The chemical stability of Mag2 and peptides 1-2 against proteinase K. The peptides were treated with proteinase K (0.002% v/v) for indicating time (h), and the amount of undegraded peptides was measured by HPLC. The digestion assay were performed in two independent experiments and each data were shown as asterisks. The data of Mag2 was taken from ref13. Peptide concentration: 1 mM.

Line 249: still needs correcting: please change “RBCs solution” to “RBCs suspension”.

Answer:

In the accordance with reviewer’s comment, the word was corrected in line 247.

Reviewer 2 Report

None.

Author Response

Thank you for reviewing our manuscript.